# Regression Trees Know Calculus

## Abstract

Regression trees have emerged as a preeminent tool for solving real-world regression problems due to their ability to deal with nonlinearities, interaction effects and sharp discontinuities. In this article, we rather study regression trees applied to well-behaved, differentiable functions, and determine the relationship between node parameters and the local gradient of the function being approximated. We find a simple estimate of the gradient which can be efficiently computed using quantities exposed by popular tree learning libraries. This allows tools developed in the context of differentiable algorithms, like neural nets and Gaussian processes, to be deployed to tree-based models. To demonstrate this, we study measures of model sensitivity defined in terms of integrals of gradients and demonstrate how to compute them for regression trees using the proposed gradient estimates. Quantitative and qualitative numerical experiments reveal the capability of gradients estimated by regression trees to improve predictive analysis, solve tasks in uncertainty quantification, and provide interpretation of model behavior.

## 1. Introduction

Tree-based methods, such as regression trees, are a workhorse of the contemporary data scientist. Their ease of use, computational efficiency and predictive capability without the need for extensive feature engineering makes them popular with practitioners. The most widely used version of regression trees approximate with greedily constructed, piecewise-constant functions than can handle data which exhibit discontinuities or divergent behavior in various parts of the feature-space. Perhaps because of their capability to tackle pathological problems, it seems that some of their

[1]Anonymous Institution, Anonymous City, Anonymous Region, Anonymous Country. Correspondence to: Anonymous Author <anon.email@domain.com>.

Preliminary work. Under review by the International Conference on Machine Learning (ICML). Do not distribute.

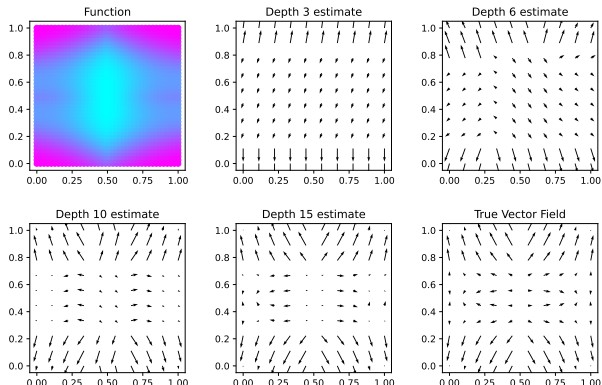

*Figure 1.* **Illustration of Gradient Estimates**. The top left gives a target function, and the bottom right gives its vector field. Shown in between are estimates of the gradient extracted from a regression tree fit to data from the function converging to the true vector field.

properties in approximating well-behaved functions may have gone unnoticed.

In this article, we will study the approximation of a continuously differentiable function $f$ on the unit cube in dimension $P$ with a piecewise constant regression tree. In particular, we will investigate a means of approximating $\nabla f$ using only information contained in the tree structure computable in a single pass through the tree. We find a simple and easily computable quantity analogous to a finite difference and, via the tree's structure, can use it to efficiently form estimates of integro-differential quantities. Previously, gradient estimation in regression trees has been studied in the context where the leaves have differentiable models, and the gradients of these models are used to estimate the gradient (e.g. Chaudhuri et al. (1995); Loh (2011)). However, the constant-leaf tree remains prevalent in practice, and the purpose of this article is rather to examine the *implicit* gradient estimation that occurs within these constant-leaf trees where, formally, the gradient of the tree is almost-everywhere zero.

With a gradient estimator in hand, we unlock for tree-based models the stable of existing gradient-based methods for variable interpretation and dimension reduction developed in other areas. Among many possibilities, we will study in particular the Active Subspace Method (Constantine, 2015), a global dimension reduction technique from the Uncer-

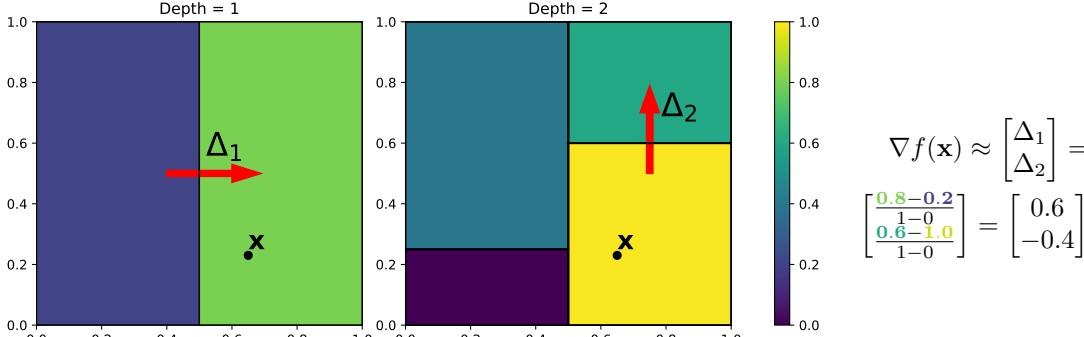

*Figure 2.* **Extract finite difference gradient approximations from a regression tree by comparing values of adjacent nodes in splits.**

tainty Quantification literature, and the Integrated Gradient Method (Sundararajan et al., 2017), a local model interpretation technique from the neural network literature.

Active Subspaces provide for linear dimension reduction, which is already commonly used in the setting of regressions trees, such as when using random projection or PCA for rotated trees (Breiman, 2001; Rodriguez et al., 2006). In contrast to these methods, active subspaces consists of supervised linear dimension reduction which takes the relationship between features and response into account. Already, supervised linear dimension reduction has been proposed for use with tree based methods where, e.g. a kernel method is used to perform the dimension reduction which is then applied to a tree-based method (Shan et al., 2015). But here, we show how to actually use the tree itself to perform a linear sensitivity analysis, rather than relying on a helper model to do this. This is essential if the analyst is interested in *model*-interpretation (as contrasted with *data*-interpretation (Chen et al., 2020)), and to the best of our knowledge the application of the Active Subspace method, enabled by our novel gradient estimates, is the first such linear sensitivity metric for trees. And while we've so far discussed what active subspaces can do for regression trees, we don't think this new relationship will be one-sided. Our numerical experiments show that regression trees can serve as scalable estimators of the active subspace, favorable to existing methods in certain circumstances. We hope this can highlight the potential for regression trees in the gradient-based UQ space.

We view the major contributions of our article as follows:

1. We develop a simple algorithm to extract gradient and integrated gradient information from regression trees.

2. We show how to use this to port gradient-based interpretability techniques from other fields to benefit interpretability of regression trees.

3. We find that in certain circumstances gradient-enabled

regression trees can produce better estimates of active subspaces than existing UQ methods.

We begin in Section 2 by discussing pertinent background on regression trees and gradient-based interpretability. Our main methodological contributions are given in Section 3 and developed theoretically in Section 4. Subsequently, we illustrate these procedure on numerical examples in Section 5 before offering conclusions and future research directions in Section 6.

## 2. Background

While small decision trees are easily interpretable, they become significantly less so as they grow in size or if they are aggregated via Random Forests (Breiman, 2001; 2002) or gradient boosting (Friedman, 2001). Consequently, developing methods for increasing the interpretability of trees is of great importance (Louppe, 2014).

### 2.1. Variable Interpretation and Selection in Regression Trees

Several methods for variable importance were suggested alongside some of the original tree-based methods (Breiman, 2017; 2001; 2002), the two most salient for our study being the Mean Decrease in Impurity (called MDI or TreeWeight (Kazemitabar et al., 2017)) and the Feature Permutation method. MDI assigns importance to variables in proportion to the mean reduction in cost when splitting along a given variable, an intuitively reasonable approach. However, MDI as originally proposed was sensitive to properties of the feature variables as well as to the depth within the tree that a split occurred, leading to "debiased" variants (Sandri & Zuccolotto, 2008; Li et al., 2019). Kazemitabar et al. (2017); Klusowski & Tian (2020) studied theoretical properties of a simplified version of this metric using only the first split ("stumps"). On the other hand Feature Permutation is based on measuring decrease in performance when shuffling a

given feature, and though the method does have some desirable properties (Ishwaran, 2007), it has come under criticism for undesirable behavior in the context of correlated predictors (Hooker et al., 2021). Empirical studies have found issues in both of these methods (Strobl et al., 2007; 2008). Against this backdrop of negative results on the behavior of these initially suggested methods came contributions in general-purpose machine learning interpretability methods. SHAP values (Lundberg & Lee, 2017) have been proposed as a general-purpose tool for understanding machine learning models. However, they would prove especially popular for understanding tree-based methods where efficient algorithms have been proposed for estimating these otherwise combinatorially difficult quantities (Lundberg et al., 2019; Karczmarz et al., 2022).

**2.2. Gradient-based Model Interpretation**

The gradient of a function tells us how its output is related to its input over short distances, and is a natural candidate to help explain the behavior of a model. Hechtlinger (2016) suggested to simply look at the gradient of a model evaluated at a particular observation to gain local understanding. Another approach is the Integrated Gradient (Sundararajan et al., 2017) introduced in the context of neural networks, which involves involves privileging some setting of input features which is called the *reference point*, which we'll denote $\mathbf{x}^*$. Subsequently, in order to explain a prediction at a given point $\mathbf{x}$, we integrate the gradient along the path from $\mathbf{x}^*$ to $\mathbf{x}$ and multiply elementwise against that difference; that is, denoting the output of the neural network as $f$:

$$IG(\mathbf{x}) := (\mathbf{x} - \mathbf{x}^*) \odot \int_{\alpha=0}^{1} \nabla f(\alpha \mathbf{x} + (1-\alpha)\mathbf{x}^*) d\alpha \,. \quad (1)$$

This theoretically appealing approach has also seen practical success in applications including medicine (Sayres et al., 2019) and chemistry (McCloskey et al., 2019).

One could also integrate the gradient over a larger part of the space to perform a more global sensitivity analysis. This is the idea behind the Active Subspace Method (Constantine, 2015). In our context, this involves defining the Active Subspace Matrix as:

$$\mathbf{C}_{\mu}^{f} := \int_{[0,1]^P} \nabla f(\mathbf{x}) \nabla f(\mathbf{x})^{\top} d\mu(\mathbf{x}) \,, \quad (2)$$

where $\mu$ is a measure. Eigenanalysis on $\mathbf{C}_{\mu}^{f}$ reveals linear combinations of features which are important, similar to PCA. Various techniques have been developed to estimate active subspaces in the presence only of input-output data. A Polynomial Ridge Approximation (PRA) procedure (Hokanson & Constantine, 2018) is available, as is a closed form estimate using Gaussian processes (GP; Wycoff et al. (2021)), and also an approach based on neural net-

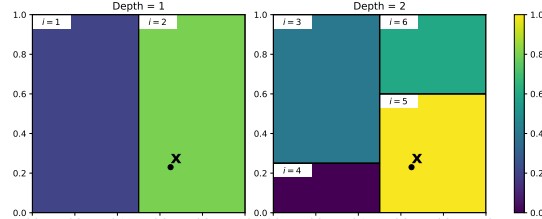

*Figure 3.* **Illustration of Notation.** In the same example tree as Figure 2 with a depth $K = 2$, examples of our notation is as follows: the indices of nodes at each depth are $\mathcal{D}_1 = \{1, 2\}$ and $\mathcal{D}_2 = \{3, 4, 5, 6\}$; the children of node 2 are $c_2^l = 5$ and $c_2^r = 6$; the bounds of node 5 are $\mathbf{l} = [0.5, 0]$ and $\mathbf{u} = [1, 0.6]$; the value of intermediate node 2 is $v_2 = 0.8$ and the value of leaf node 6 is 0.6; since the "root node" 0 is split along the x-axis, $\sigma_0 = 1$ and since nodes 1 and 2 are split along the y-axis, $\sigma_1 = \sigma_2 = 2$; since the point $\mathbf{x}$ lies within the nodes 2 and 5 at depths 1 and 2 respectively, we have that $B^1(\mathbf{x}) = 2$ and $B^2(\mathbf{x}) = 5$.

works called the Deep Active Subspace Method (Tripathy & Bilionis, 2019; Edeling, 2023, DASM) has been proposed.

# 3. Integro-Differentiation of Regression Trees

In this section we propose an easy to compute estimator of derivatives and integrals of derivatives for regression trees. While the underlying idea in this article is simple, discussing it rigorously requires a fair amount of notation; Figure 3 illustrates our notation on simple regression tree. Say we fit a regression tree of depth $K$ to data $(\mathbf{x}_n, y_n = f(\mathbf{x}_n) + \epsilon_n)$ with $\mathbf{x}_n \in [0, 1]^P, y_n \in \mathbb{R}$ for $n \in \{1, \dots, N\}$, and $\epsilon_n$ mean zero with finite variance. Purely for simplicity, we will assume that the tree has leaf nodes only at depth $K$, i.e. there is no path through the tree that ends before depth $K$. Associate with every node from all depths an integer index $i$ such that the root node is labeled 0 and if node $i$ is deeper in the tree than node $j$, then $i > j$. That is, the indices at depth 1 are given by $\{1, 2\}$, at depth 2 by $\{3, 4, 5, 6\}$, and generally at depth $k$ the indices are given by $\mathcal{N}_k = \{2^{k-1} + 1, \dots, 2^k\}$. We denote the variable along which the $i$th node splits by $\sigma_i \in \{1, \dots, P\}$ and the threshold at which that split occurs as $\tau_i$.

## 3.1. Differentiation

The difference between the mean response $\mu_i^l$ on the left side of node $i$'s split (containing points in that node where $x_{\sigma_i} \leq \tau_i$) and the mean response $\mu_i^r$ on the right side (where $x_{\sigma_i} > \tau_i$) contains information about the $\sigma_i^{\text{th}}$ component of the gradient in that area. By dividing this difference proportionally to the size of the node along dimension $\sigma_i$, we form a quantity similar to a finite difference. Let $[l_p^i, u_p^i]$ denote the extent of node $i$ along dimension $p$ for $p \in$

$\{1, \ldots, P\}$. Then define the following quantity:

$$\left.\frac{\tilde{\partial} f}{\tilde{\partial} x_{\sigma_i}}\right|_i := \frac{2(\mu_i^r - \mu_i^l)}{u_{\sigma_i}^i - l_{\sigma_i}^i} \approx \frac{\partial f}{\partial x_{\sigma_i}}(\mathbf{x}) \ \forall \mathbf{x} \in [\mathbf{l}^i, \mathbf{u}^i]. \quad (3)$$

We have used the notation $\mathbf{l}^i = [l_1^i, \ldots, l_P^i]$ to denote the vector of lower bounds of the extent of node $i$ and similar for $\mathbf{u}^i$, and by $[\mathbf{l}^i, \mathbf{u}^i]$ we denote the set of points lying within the bounds of the $i^{\text{th}}$ node. The ratio $\left.\frac{\tilde{\partial} f}{\tilde{\partial} x_{\sigma_i}}\right|_i$ contains information about the partial derivative of $f$ with respect to $x_{\sigma_i}$ (which is, again, the variable that node $i$ splits along) inside of $[\mathbf{l}^i, \mathbf{u}^i]$. The idea that $\left.\frac{\tilde{\partial} f}{\tilde{\partial} x_{\sigma_i}}\right|_i \approx \left.\frac{\partial f(\mathbf{x})}{\partial x_{\sigma_i}}\right|_i$ for any $\mathbf{x}$ in node $i$ only makes sense if the gradient does not vary much within it. We expect this to be the case only for very small nodes, or in other words, very deep trees, estimating functions with gradients which vary smoothly; we make this precise in the next section.

At a particular node, we can only form estimates of a single partial derivative. However, if our tree is sufficiently deep, we can combine estimates of partial derivatives from nodes at multiple depths to form an estimate of the entire gradient. Recall that $\mathcal{N}_k = \{2^{k-1} + 1, \ldots, 2^k\}$ contains the index of all nodes of depth $k$ and let $\rho_i$ denote the index of the parent of node $i$ (which will be a member of $\mathcal{N}_{k-1}$). Then Algorithm 1 shows how to aggregate the partial derivative estimates into a gradient estimate.

---

**Algorithm 1** Tree-Based Gradient Estimation

$\mathbf{G}^i \leftarrow \mathbf{0} \in \mathbb{R}^P$ for all $i$.
**for** $k \in \{1, \ldots, K\}$ **do**
  **for** $i \in \mathcal{N}_k$ **do**
    $\mathbf{G}^i \leftarrow \mathbf{G}^{\rho_i}$ {Get parent's estimate}
    $\mathbf{G}^i[\sigma_i] \leftarrow \frac{2(\mu_i^r - \mu_i^l)}{u_{i,\sigma_i} - l_{i,\sigma_i}}$ {Update along split direction}
  **end for**
**end for**

---

We thus have associated each node $i$ with an estimator of the gradient, though if we have not split along a variable $p$ in the tree upstream of $i$, the estimate of that component will simply be 0. This is actually somewhat reasonable, as the tree will split along variables with small gradient components less frequently.

Of course, nothing would stop us from comparing values of means that are adjacent in the input domain but not adjacent in the tree structure (for example, nodes 3 and 6 of Figure 3). However, the advantage of considering only nodes adjacent in the tree is that this Tree-Based Gradient Estimator (TBGE) can be computed efficiently merely by traversing the tree. Furthermore, if we needed only an estimate of a gradient at a particular point $\mathbf{x}$, we would need only to traverse the regression tree in the standard manner, visiting only nodes upstream of the leaf nod $\mathbf{x}$. To be explicit, let

$B(\mathbf{x})$ be the index of the node of the greatest depth which contains $\mathbf{x}$. We use as our estimate of $\nabla f(\mathbf{x})$ the TBGE associated that node; that is, $\tilde{\nabla} f(\mathbf{x}) = \mathbf{G}^{B(\mathbf{x})}$.

We can see that once each feature has been iterated over once, $\mathbf{G}_i$ will be in some sense an estimator of the gradient within node $i$. Furthermore, in the large data limit, the analysis of Appendix A.2 reveals that this will occur within $(P - 1) \log_2 \left( \frac{\max_p |\frac{\partial f(x)}{\partial x_p}|}{\min_p |\frac{\partial f(x)}{\partial x_p}|} \right)$ splits when approximating linear functions. This suggests that depth should grow linearly with dimension, or least with the number of nonzero coefficients in the gradient.

### 3.2. Integration

We next develop estimators for quantities of the form:

$$\mathcal{I}(f) = \int h(\nabla f(\mathbf{x})) d\mu(\mathbf{x}). \quad (4)$$

Here, $h$ is some integrable $\mathbb{R}^P \to \mathcal{H}$ function and $\mu$ is a measure on $\mathbf{R}^P$. The Integrated Gradient at point $\mathbf{z}$ with reference point $\mathbf{z}^*$ may be written in this form by choosing $h(\mathbf{a}) = (\mathbf{z} - \mathbf{z}^*) \odot \mathbf{a}$, $\mathcal{H} = \mathbf{R}^P$, and $\mu$ as the degenerate uniform measure on the line segment between $\mathbf{z}$ and $\mathbf{z}^*$. Similarly, Active Subspace can also be seen to belong to this class by setting $h(\mathbf{a}) = \mathbf{a}\mathbf{a}^\top$, $\mathcal{H} = \mathbb{R}^{P \times P}$ and arbitrary $\mu$.

We will consider two classes of estimators for such quantities. The first is a Monte Carlo Estimator (MCE) which is appropriate whenever $\mu$ is a probability measure which it is possible to sample from. The second is a Partition-Based Estimator (PBE) which is suitable whenever we can compute $\mu([\mathbf{a}, \mathbf{b}])$ for arbitrary $\mathbf{a}, \mathbf{b}$.

Start with the MCE. We fix some Monte Carlo sample size $M$ and form the standard Monte Carlo approximation, then plug in the TBGE where the gradient appears:

$$\mathcal{I}(f) \approx \frac{1}{M} \sum_{\mathbf{x}_m \sim \mu} h(\nabla f(\mathbf{x}_m)) \approx \frac{1}{M} \sum_{\mathbf{x}_m \sim \mu} h(\mathbf{G}^{\mathbf{x}_m}). \quad (5)$$

We denote this estimator by $\hat{\mathcal{I}}_{MC}(f)$. In particular, we can form a Tree-Based Integrated Gradient (TBIG) as follows by sampling random uniform variables $u_m$ on $[0, 1]$:

$$\hat{IG}(\mathbf{x}) = (\mathbf{x} - \mathbf{x}^*) \odot \frac{1}{M} \sum_{m=1}^{M} \mathbf{G}^{B(u_m \mathbf{x} + (1 - u_m)\mathbf{x}^*)}. \quad (6)$$

Next, we consider the PBE. This involves simply computing a weighted average of the action of $h$ on the TBGE on each node at the penultimate depth $K - 1$, weighted by the measure of that node. For simplicity, we assume that $\mu([0, 1]^P) = 1$. In notation, we define the estimator as

$$\hat{\mathcal{I}}_{PE}(f) = \sum_{i \in \mathcal{N}_{K-1}} h(\mathbf{G}^i) \mu([\mathbf{l}^i, \mathbf{u}^i]). \quad (7)$$

We can use this to form Tree-Based Active Subspace (TBAS) for $f$ as follows:

$$\hat{C}_\mu^f = \sum_{i \in \mathcal{N}_{K-1}} \mathbf{G}^i \mathbf{G}^{i\top} \mu([\mathbf{l}^i, \mathbf{u}^i]).$$ (8)

Which estimator is preferred in practice will depend on which properties of $\mu$ are computationally tractable. If both sampling and computing measure of rectangles is simple for $\mu$, both will be available, and in this circumstance, the PBE will be favorable as it does not have Monte Carlo error.

## 4. Asymptotic Analysis

We'll now make rigorous the intuition developed in the previous section with a basic asymptotic analysis of the TBGE under the assumption that the mean value in a node converges at the usual square root rate, which is the case under typical regularity conditions.

**Theorem 4.1.** *Let $S(N)$ denote the least number of splits in the tree along any variable as a function of the sample size. Under Assumption A.1 of the Appendix, we have that:*

$$\frac{\tilde{\partial} f(\mathbf{x})}{\tilde{\partial} x_{\sigma_i}} = \frac{\partial f(\mathbf{x})}{\partial x_{\sigma_i}} + O_P\left(\sqrt{\frac{S(N)}{N}} + 2^{-\frac{S(N)}{P}}\right).$$

*Proof.* We give only an outline of the proof, with the details given in the Appendix. Expanding $f$ around $x$ can be used to show that

$$\frac{\tilde{\partial} f(\mathbf{x})}{\tilde{\partial} x_p} = \frac{\partial f(\mathbf{x})}{\partial x_{\sigma_i}} + O_P\left(\frac{1}{\sqrt{\eta(N)}}\right) + o(\operatorname{diam}([\mathbf{l}, \mathbf{u}])),$$

where $\eta(N)$ is the least number of observations in any node. Then, noting that $\eta(N) \leq \frac{N}{S(N)}$ and measuring the decrease in node width with depth yields the desired result. □

This shows us the trade-off between decreasing the variance of the mean estimate in each cell and narrowing the sizes of the cells. Taking $S(N) = P \log N$ yields a convergence rate of $\sqrt{PN^{-1} \log N}$.

In the remainder of this section, we will establish consistency of estimates for integro-differential quantities, which will require only consistency of the TBGE. We see that this occurs whenever $S(N)$ grows to $\infty$ sublinearly such that there are eventually arbitrarily many splits, each with arbitrarily many observations as the sample size increases. We begin with the MCE; the remaining proofs are in the Appendix.

**Theorem 4.2.** *Under Assumptions A.1 and A.2 of the appendix, the Partition-Based estimator converges to the true*

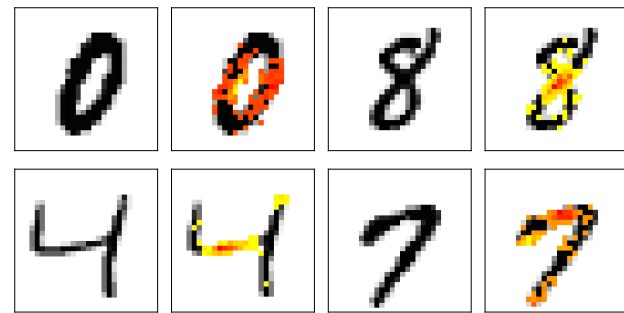

*Figure 4.* **Integrated Gradient for Trees.** Each pair of panels gives a training example from MNIST. The second pair in the image superimposes the IG values onto the example. Redder means more strongly suggesting correct class membership.

*integro-differential quantity as the sample size diverges. That is,*

$$\lim_{N \to \infty} \hat{\mathcal{I}}_{PB}(f) = \mathcal{I}(f).$$

*where $\mathcal{I}(f)$ is the integro-differential quantity given in Equation 4.*

Theorem A.1 of the Appendix gives a similar result for the MCE.

We also explicitly state there the implications of these results for Active Subspace and Integrated Gradient estimation, namely the consistency of the estimators $\hat{C}_\mu^f$ and $\hat{IG}$ proposed in Section 3.

We have thus established consistency of gradient-based model interpretation quantities for regression trees. Next, we will empirically study their behavior in finite samples via a battery of numerical experiments.

## 5. Numerical Experiments

We now study how the proposed gradient estimator might be profitably exploited in practice. We begin with a qualitative study showing how a tree-based integrated gradient (TBIG) can facilitate local model interpretation. Then come three quantitative studies, first investigating the potential of a Tree-Based Active Subspace (TBAS) to improve prediction accuracy of a downstream tree via a rotation of the space. Subsequently, we evaluate the capacity of regression trees to estimate the Active Subspace in low and high dimension. Finally, we end with another qualitative study demonstrating how a TBAS can provide data visualization.

### 5.1. Integrated Gradient for Tree-Based Methods

We now consider random forest classifier fit the to two subsets of the MNIST dataset, one contrasting zeros against

| Dataset: | bike | concrete | gas | grid | keggu | kin40k | obesity | supercond |
|---|---|---|---|---|---|---|---|---|
| **Regression Tree (Depth 4)** | | | | | | | | |
| TBAS | **0.635** | **0.47** | **0.578** | **0.688** | **0.194** | **0.856** | **0.128** | **0.511** |
| Id | **0.655** | 0.537 | 0.595 | 0.773 | 0.35 | 0.963 | 0.226 | **0.514** |
| PCA | **0.655** | 0.543 | **0.591** | 0.773 | 0.349 | 0.964 | 0.226 | **0.513** |
| Rand | 0.656 | 0.524 | 0.593 | 0.752 | 0.349 | 0.95 | 0.226 | **0.513** |
| **Regression Tree (Depth 8)** | | | | | | | | |
| TBAS | **0.402** | **0.35** | **0.429** | **0.521** | **0.078** | **0.586** | **0.094** | **0.392** |
| Id | **0.405** | 0.403 | **0.432** | **0.522** | 0.121 | 0.862 | **0.103** | **0.395** |
| PCA | **0.407** | **0.395** | **0.423** | **0.524** | 0.122 | 0.872 | **0.107** | **0.392** |
| Rand | **0.411** | **0.391** | **0.435** | 0.55 | 0.124 | 0.836 | **0.109** | **0.397** |
| **Random Forest (Depth 4)** | | | | | | | | |
| TBAS | **0.609** | **0.406** | **0.559** | **0.602** | **0.161** | **0.802** | **0.123** | **0.479** |
| Id | 0.65 | 0.462 | **0.574** | 0.659 | 0.344 | 0.954 | 0.173 | **0.485** |
| PCA | 0.649 | 0.462 | **0.567** | 0.654 | 0.344 | 0.954 | 0.174 | **0.486** |
| Rand | 0.646 | 0.458 | **0.57** | 0.66 | 0.33 | 0.938 | 0.173 | **0.486** |

*Table 1.* **Predictive Impact of Various Transformations on Selected Datasets**. Numbers give 100-fold RMSE; bold indicates confidence interval overlaps with the lowest confidence interval.

eights and the other fours against sevens. While classification was not the focus of this article, we present some preliminary numerical experiments in Appendix C. We use the estimator $\hat{IG}$ of Equation 6 with $M = 500$ random points along a given path. Figure 4 gives the results of this analysis. We see that when comparing eights against zeros, the intersection point of the eight is most important, while the left and right sides of the zero are. This makes sense as they are the parts of each digit which are distinct. By contrast, the shared upper and lower arcs are not highlighted. In the four versus seven case, the location of the horizontal bar seems to be most influential in determining one versus the other. This suggests that the model is comparing the relative position of the bar to the rest of the digit in making its classifications.

### 5.2. Active-subspace Rotated Trees

This section evaluates the ability of TBAS to improve the accuracy of a downstream predictive analysis. Given some mapping $\mathbf{L}$, we refer to postmultiplication of the feature matrix $\mathbf{X}$ to form a new feature matrix $\mathbf{Z} := \mathbf{XL}$ as a rotation. In standard tree-based methods, non-diagonal rotations can have an impact on predictive performance because the axes along which splits are made have been changed. In order to quantitatively assess the utility of TBAS, we compare the prediction error of regression trees and random forests on eight benchmark datasets fit on data augmented by a rotation of the space. We compare TBAS rotations to the those made by drawing orthogonal directions uniformly over the Grassmannian (Rand; similar to Breiman (2001)) as well as

those formed from a Principal Components Analysis (PCA; similar to Rodriguez et al. (2006)) and no rotation at all (denoted Id). Table 1 presents the results of this analysis. TBAS does at least as well as the other methods, and sometimes offers significant improvement (such as on the Kin40k and Keggu datasets).

### 5.3. Computationally Efficient Active Subspace Estimation in Low Dimension

In this section we demonstrate the capability of TBAS to offer extremely fast, gradient-free estimation of the active subspace in low dimension when significant data are available. We compare to the three popular methods for gradient-free active subspace estimation introduced in Section 2.2, based on a Gaussian Process (GP), Ridge Polynomial (PRA), and Neural Network (DASM). Each of these methods was tasked to estimate a one dimensional active subspace in dimensions 2, 3 and 4, using a logarithmically spaced grid of sample sizes ranging from 10 to 10,000 on a toy algebraic function. We found that the GP and PRA methods worked efficiently when sample sizes were small, but computation time grew quickly, such that we were only able to run these methods for samples of size 150 or less. Figure 5 shows the active subspace error against the execution time. We see that the tree-based estimator forms the majority of the Pareto front in all three cases.

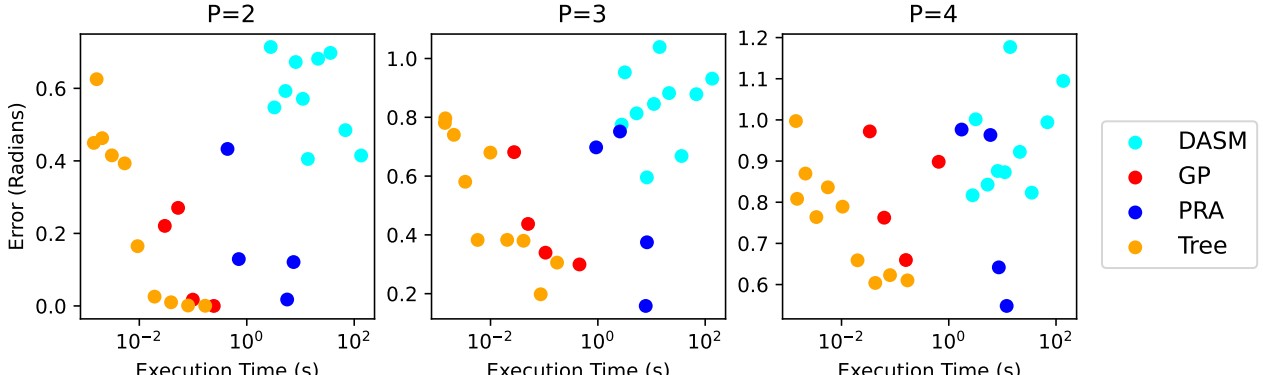

*Figure 5.* **Active Subspace Estimation in Low Dimension.** Execution time (x-axis) and Subspace Estimation Error (y-axis) for the four methods, lower is better.

### 5.4. Estimating Sparse Active Subspaces in High Dimension

Imposing sparsity in the coefficients of linear dimension reduction can improve interpretability (Zou et al., 2006). Because of the manner that tree-based methods produce gradient estimates, we conjecture that TBAS has built-in inductive bias to favor entry-wise active subspaces. Our analysis in Appendix A.2 revealed that in the large sample limit, the tree-based gradient estimator will behave as though it was in a lower dimensional space, with dimension given by the cardinality of the gradient. In this section, we will investigate this property by comparing TBAS against the DASM in estimating a sparse active subspace in dimensions 10, 50 and 100. We will use the same setup as the previous section, except that the true subspace has only three nonzero coordinates. Because of the sample size that is required to estimate an active subspace in high dimension, it is computationally infeasible to deploy the GP or PRA active subspace estimation methods. Figure 6 shows the results of this analysis. The TBAS estimator is able to achieve much better accuracy with a significantly smaller cost,

### 5.5. Dimension Reduction with the Active Subspace

We now turn to the qualitative benefits of performing an active subspace analysis using TBAS. We used the NHEFS dataset of biochemistry tape and mortality data which were used by Lundberg et al. (2019). This consisted of a dataset of 14,407 observations and 90 complete variables. We fit a TBAS to this dataset using a regression tree of depth 15, requiring at least 10 samples per leaf. We subsequently performed an eigendecomposition of the estimated active subspace matrix. Like Lundberg et al. (2019), we found that age was the most important variable and it mapped cleanly onto the first active subspace dimension. However, we found that the next two dimensions consisted of many

variables (see Appendix B.3). Figure 7 shows there result of this analysis, which reveals that after Age there appears to be a gap in the spectrum of the active subspace matrix (Constantine, 2015), indicating the presence of an active subspace of dimension two (left panel). The middle panel shows a projection of the data, which breaks into two clusters along the second principal component, while the right panel shows a projection of randomly sampled points to visualize the predictive surface of the function. The right panel reveals the predictive surface has a fairly simple, "S-shaped" form over these two dimension. It would not be possible to detect this by considering only main effects or two factor interactions.

## 6. Discussion

**Summary:** We proposed a simple method for estimating gradients in sufficiently deep regression trees on large datasets. Subsequently, we demonstrate how these estimates could be used to calculate active subspaces and integrated gradients. We found significant improvement in predictive performance could be achieved by using the tree-based active subspace to rotate the space, and also found that these estimates executed quicker than existing gradient-free active subspace estimators and had useful inductive bias towards detecting coordinate-sparse subspaces. Also, we were not able to run the PRA and GP based active subspace estimators on larger datasets due to computational constraints. Finally, we used these gradient estimates to reveal the simple global structure of a tree-based model fit to a complex dataset as well as the mechanism by which a random forest conducted MNIST classification.

**Limitations:** When it comes to limitations of our experiments, when comparing our active subspace estimates to the DASM, we used a particular neural network architecture and optimizer. It is possible that a different configuration

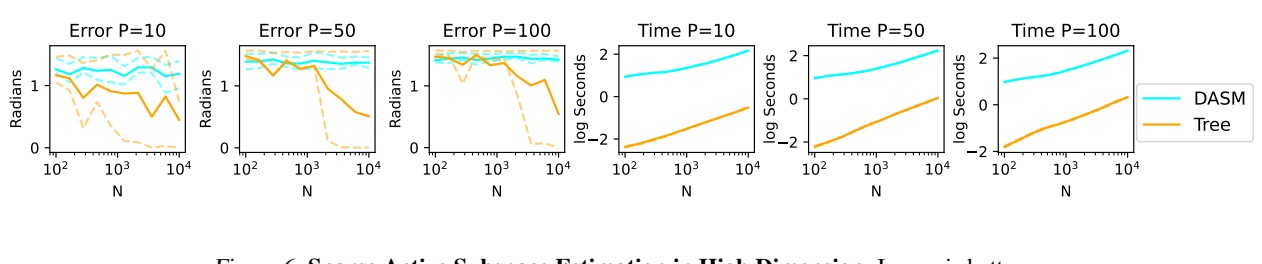

*Figure 6.* **Sparse Active Subspace Estimation in High Dimension.** Lower is better.

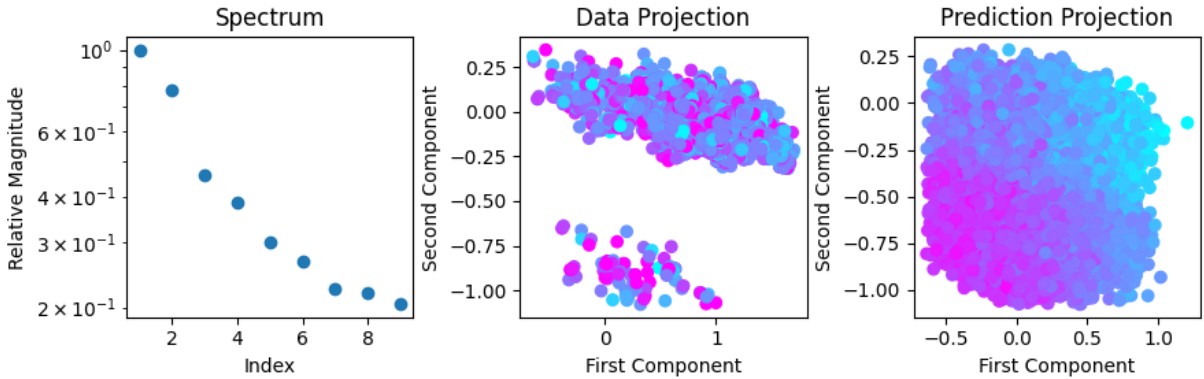

*Figure 7.* **Projection with TBAS.** *Left:* Spectrum of active subspace matrix suggests a 2D active subspace. *Middle:* Projection of data onto active subspace. *Left:* Projection of predictive surface.

would have performed better. Additionally, when it comes to limitations of the method, our analysis suggests that it would take a very deep regression tree to estimate gradients in high dimension if the gradient is dense in all of its entries.

**Conclusions:** We think this work offers two high level conclusions. First, that trees may have a place in the field of Uncertainty Quantification, which has more commonly used differentiable but slow to estimate surrogates such as Gaussian processes and neural networks. Secondly, we hope that this work can also enable improve cross-pollination between developments in interpretability for neural networks and for regression trees by creating analogs for gradient-based neural network techniques.

We also would like to comment on how TBAS fit into the existing literature on interpretability in trees. It is somewhat distinct from the axiomatic approach to interpretability proffered by SHAP (Lundberg & Lee, 2017) and Integrated Gradients (Sundararajan et al., 2017). While an axiomatic approach can be useful, its universality should not be exaggerated. Hancox-Li & Kumar (2021) write about how it is unlikely that any variable selection method could satisfy all possible demands, no matter how attractive its axiomatic foundation. Since the active subspace is parameterized by a measure, it actually consists of an entire family of analyses, with different choices of emphasis over the input space possibly leading to different conclusions.

**Future Work:** We are excited about the future work opened up by the here-proposed gradient estimates. First, while we have conducted some preliminary analyses in Appendix C, more work is needed to understand the properties of these estimates in classification problems with categorical inputs and missing data. We are also interested in the possibility of estimating higher order derivatives from tree structure. Furthermore, extending the class of functions approximated to nondifferentiable functions is of interest: does the quantity proposed here converge in such a case, and to what? But we are most excited about the potential to bring in other gradient-based techniques to tree-based methods. For instance, physics-informed machine learning (e.g. Raissi et al. (2019; 2017)) has been making waves in the fluid dynamics community (Cai et al., 2021) among other applications, where they allow the analyst to use information about function derivatives to improve predictions of differentiable models such as neural networks or Gaussian processes. Our gradient estimator opens up the possibility of deploying this technique in the context of regression trees.

## Impact Statement

This paper presents work whose goal is to advance the field of Machine Learning. There are many potential societal consequences of our work, none which we feel must be specifically highlighted here.

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

## A. Proofs

We begin by stating our required assumptions. Assumption 1 will be required for Theorem A, establishing the convergence rate of our gradient estimator. The other results, regarding consistency of integro-differential estimators, will require Assumption 2 as well.

**Assumption 1.** *Let $y_n = f(\mathbf{x}_n) + \epsilon_n$. We assume that:*

1. *The input points $\mathbf{x}_n$ are sampled independently from a continuous distribution on $[0, 1]^P$.*

2. *The estimate of the mean of $f(\mathbf{x})$ over some interval $[\mathbf{a}, \mathbf{b}]$ converges at usual square root rate; this is achieved for example if the error terms $\epsilon_1, \ldots, \epsilon_N$ are such that $\mathbb{E}[\epsilon_n] = 0$; $\mathbb{V}[\epsilon_n] \leq \infty$ and $\epsilon_{n_1} \perp\!\!\!\perp \epsilon_{n_2} \forall n_1, n_2 \in \{1, \ldots, N\}$.*

3. *We alternate between which variable is split along at each depth.*

4. *$f$ is continuously differentiable.*

Items 1 and 2 are necessary for the mean estimates in leaf nodes to converge to the mean value of $f$ on that node, while 3 and 4 are necessary to make a series of $f$ within a node accurate.

**Assumption 2.** *Let $S(N)$ denote the depth as a function of sample size.*

1. *$\lim_{N \to \infty} S(N) = \infty$.*

2. *$\lim_{N \to \infty} \frac{S(N)}{N} = 0$.*

3. *$h$ is measurable and bounded.*

Assumptions 1 and 2 ensure that we accumulate enough data in each leaf node, while still accumulating sufficiently small leaf nodes, to achieve gradient consistency.

We are now prepared to prove our results.

**Theorem 4.1.** *Let $S(N)$ denote the least number of splits in the tree along any variable as a function of the sample size. Under Assumption A, we have that:*

$$\frac{\tilde{\partial} f(\mathbf{x})}{\tilde{\partial} x_{\sigma_i}} = \frac{\partial f(\mathbf{x})}{\partial x_{\sigma_i}} + O_P\left(\sqrt{\frac{S(N)}{N}} + 2^{-\frac{S(N)}{P}}\right).$$

*Proof.* If $\mathbf{x} \in [\mathbf{l}, \mathbf{u}]$, expand $f$ around $\mathbf{u}$ (or any other point in $[\mathbf{l}, \mathbf{u}]$):

$$f(\mathbf{x}) = f(\mathbf{u}) + \nabla f(\mathbf{u})^\top (\mathbf{x} - \mathbf{u}) + o(\|\mathbf{x} - \mathbf{u}\|) \tag{9}$$
$$= f(\mathbf{u}) + \nabla f(\mathbf{u})^\top (\mathbf{x} - \mathbf{u}) + o(\text{diam}([\mathbf{l}, \mathbf{u}])), \tag{10}$$

where $\text{diam}(\mathcal{A})$ is the diameter of the set $\mathcal{A}$ and is given by $\max_{\mathbf{x}_1, \mathbf{x}_2 \in \mathcal{A}} \|\mathbf{x}_1 - \mathbf{x}_2\|$. We denote the minimal number of points in any node by $\eta(N)$. Then:

$$\frac{\tilde{\partial} f(\mathbf{x})}{\tilde{\partial} x_p} = \frac{\partial f(\mathbf{x})}{\partial x_{\sigma_i}} + O_P\left(\frac{1}{\sqrt{\eta(N)}}\right) + o(\text{diam}([\mathbf{l}, \mathbf{u}])). \tag{11}$$

Denote by $S(N)$ the depth of the tree as a function of the sample size. If we are making cuts at midpoints in each node, then $\text{diam} = \max_{1 \leq p \leq P} \frac{1}{2^{S_p}}$, where $S_p$ is the number of splits along variable $p$. It's clear that reducing the diameter is most speedily achieved by alternating which variable is split along such that $\text{diam} = 2^{-\lfloor \frac{S}{P} \rfloor} \leq C 2^{-\frac{S}{P}}$ for some $C$. We have that $\eta(N) \leq \frac{N}{S(N)}$ with equality if the points are evenly distributed. So $O_P(\sqrt{\frac{1}{\eta(N)}}) = O_P(\sqrt{\frac{S(N)}{N}})$ and $o(\text{diam}([\mathbf{l}, \mathbf{u}]) = o(2^{-\frac{S}{P}})$, which taken together with 11 yield the desired result. $\qquad \square$

**Theorem 4.2.** *Under Assumptions A and A, the Partition-Based estimator converges to the true integro-differential quantity as the observation sample size diverges. That is,*

$$\lim_{N \to \infty} \hat{\mathcal{I}}_{PB}(f) = \lim_{N \to \infty} \sum_{i \in \mathcal{N}_{K-1}} h(\mathbf{G}^i)\mu([\mathbf{l}^i, \mathbf{u}^i]) \,.\, = \int h(\nabla f(\mathbf{x}))d\mu(\mathbf{x}) = \mathcal{I}(f) \,.$$

*Proof.* Since the function sequence $\mathbf{g}^k(\mathbf{x}) := G_{B^k(\mathbf{x})}$ converges pointwise to $\nabla f(\mathbf{x})$ by Proposition 1, we need only establish a function $H(\mathbf{x})$ which dominates $\mathbf{g}^k(\mathbf{x})$ and apply the Dominated Convergence Theorem.

To this end, denote by $C^r, C^l$ the child nodes of node $i$ and examine its gradient estimator's $p$th entry, given by:

$$\frac{2(\mu_i^r - \mu_i^l)}{u_p^i - l_p^i} \xrightarrow{N \to \infty} \frac{2\left(\frac{1}{|\mathcal{N}_{C^r}|} \int_{\mathcal{N}_{C^r}} f(\mathbf{x})d\mathbf{x} - \frac{1}{|\mathcal{N}_{C_i^r}|} \int_{\mathcal{N}_{C_i^l}} f(\mathbf{x})d\mathbf{x}\right)}{u_p^i - l_p^i} \,. \tag{12}$$

The magnitude of this difference in averages is bounded by the magnitude of the difference of extremes:

$$\left| \frac{1}{|\mathcal{N}_{C_i^r}|} \int_{\mathcal{N}_{C_i^r}} f(\mathbf{x})d\mathbf{x} - \frac{1}{|\mathcal{N}_{C_i^r}|} \int_{\mathcal{N}_{C_i^l}} f(\mathbf{x})d\mathbf{x} \right| \leq \max_{(\mathbf{x}_1, \mathbf{x}_2) \in \mathcal{N}_{C_i^r} \times \mathcal{N}_{C_i^l}} |f(\mathbf{x}_1) - f(\mathbf{x}_2)| \,. \tag{13}$$

But since $f$ is continuously differentiable, it is also Lipschitz continuous (call the constant $L$), and we have that:

$$\max_{(\mathbf{x}_1, \mathbf{x}_2) \in \mathcal{N}_{C_i^r} \times \mathcal{N}_{C_i^l}} |f(\mathbf{x}_1) - f(\mathbf{x}_2)| \leq L\|\mathbf{x}_1 - \mathbf{x}_2\|_2 \leq PL\|\mathbf{x}_1 - \mathbf{x}_2\|_\infty \,. \tag{14}$$

Therefore:

$$\left| \frac{2(v_{C_i^r} - v_{C_i^l})}{u_p^i - l_p^i} \right| \leq \left| \frac{2(PL\|\mathbf{x}_1 - \mathbf{x}_2\|_\infty)}{u_p^i - l_p^i} \right| \leq 2PL \,. \tag{15}$$

Hence the constant $2PL$ bounds $g^k(\mathbf{x})$, and thence $4P^2L^2$ bounds the outer product function. Since the integral is over the unit hypercube, the constant function is integrable and we can apply the Dominated Convergence Theorem to yield the desired result.

$\square$

**Theorem A.1.** *Under Assumptions A and A, the Monte Carlo estimator converges to the true integro-differential quantity as the Monte Carlo sample size and the observation sample size diverge. That is,*

$$\lim_{N,M \to \infty} \hat{\mathcal{I}}_{MC}(f) = \lim_{N,M \to \infty} \frac{1}{M} \sum_{\mathbf{x}_m \sim \mu} h(\mathbf{G}^{\mathbf{x}_m}) = \int h(\nabla f(\mathbf{x}))d\mu(\mathbf{x}) = \mathcal{I}(f) \,.$$

*Proof.* This follows from the fact that

$$\lim_{N \to \infty} \lim_{M \to \infty} \frac{1}{M} \sum_{\mathbf{x}_m \sim \mu} \mathbf{G}^{B^{K(N)-1}(\mathbf{x}_m)} \mathbf{G}^{B^{K(N)-1}(\mathbf{x}_m)\top} = \tag{16}$$

$$= \lim_{N \to \infty} \int_{\mathbf{x} \in [0,1]^P} \mathbf{G}^{B^{K(N)-1}(\mathbf{x})} \mathbf{G}^{B^{K(N)-1}(\mathbf{x})\top} d\mu = \lim_{N \to \infty} \sum_{i \in \mathcal{D}_k} \mathbf{G}_i \mathbf{G}_i^\top \mu(\mathcal{N}_i) \tag{17}$$

and an application of Theorem 4.2.

$\square$

## A.1. Implications for Active Subspaces and Integrated Gradients

We conclude this section by explicitly stating the implications of these results for Tree-based Active Subspace (TBAS) estimation.

**Corollary A.2.** *Let $K(N)$ denote the depth of the regression tree as a function of $N$. Under Assumptions A and A, we have that*

$$\lim_{N \to \infty} \sum_{i \in \mathcal{D}_{K(N)-1}} \mathbf{G}_i \mathbf{G}_i^\top \mu(\mathcal{N}_i)$$

$$= \int_{[0,1]^P} \nabla f(\mathbf{x}) \nabla f(\mathbf{x}) d\mu(\mathbf{x}) .$$

*Proof.* Follows from Theorem 4.2. □

As well as for Tree-Based Integrated Gradient (TBIG) estimation.

**Corollary A.3.** *Let $K(N)$ denote the depth of the regression tree as a function of $N$. Under Assumptions A and A, we have that, assuming that $u_n$ are iid uniform on [0,1]:*

$$\lim_{k,N,M \to \infty} (\mathbf{x} - \mathbf{x}^*) \odot \frac{1}{M} \sum_{m=1}^{M} \mathbf{G}^{B^k\left(u_m \mathbf{x} + (1-u_m)\mathbf{x}^*\right)}$$

$$= (\mathbf{x} - \mathbf{x}^*) \odot \int_{\alpha=0}^{1} \nabla f(\alpha \mathbf{x} + (1-\alpha)\mathbf{x}^*) dx .$$

*Proof.* Follows from Theorem A.1. □

## A.2. Thresholding Behavior and Iteration

In this section we study large sample iteration behavior of a greedily estimated regression tree on linear functions $f(\mathbf{x}) = \mathbf{a}^\top \mathbf{x} + b$. As the sample size diverges, we have that $\mathbb{V}[\mathbf{a}^\top \mathbf{x}; [\mathbf{l}, \mathbf{u}]] = \frac{\sum_p a_p^2 (u_p^i - l_p^i)^2}{12}$ such that that optimization may be rewritten as:

$$\operatorname*{argmin}_{s \in \{1,\dots,P\}, \, t \in [l_p^i, u_p^i]} 2 \sum_{p \neq s} a_p^2 (u_p^i - l_p^i)^2 + a_s^2 \left( (u_p^i - t)^2 + (t - l_p^i)^2 \right). \tag{18}$$

For any fixed $s$, the minimum with respect to $t$ occurs at $\frac{u_{i,s} + l_{i,s}}{2}$, yielding the profile problem:

$$\operatorname*{argmin}_{s \in \{1,\dots,P\}} 2 \sum_{p \neq s} a_p^2 (u_p^i - l_p^i)^2 + \frac{a_p^2}{2} (u_s^i - l_s^i)^2 . \tag{19}$$

By comparing any two costs, we can see that the $s$ that will be chosen is that such that the quantity $|a_s|(u_s^i - l_s^i)$ is maximized.

Such is the behavior for a specific node at a specific depth. Let's now reflect on what this implies for the iteration. Initially, $u_p - l_p = 1$ for all $p$, such that the variable with the largest coefficient $p$ will be chosen for the first split; let's call that variable $p_1 := \operatorname{argmax}_p |a_p|$. Subsequently, if $p_1$ is more than twice the size of the second largest coefficient $p_2 := \operatorname{argmax}_{p \neq p_1} |a_p|$, the second split in each of the two leaves will occur along variable $p_1$ as well. Otherwise, it will occur along variable $p_2$. Under our large sample assumptions, all the nodes at a given depth will behave uniformly, splitting alternatively along variables until the terms $|a_p|(u_p^i - l_p^i)$ are within a multiple of two of one another. Denote the index of the minimum coefficient as $p^\wedge := \operatorname{argmin}_p |a_p|$ and assume for now that $a_{p^\wedge}$ is nonzero. Using the notation $\lfloor c \rfloor$ to denote the integer part of $c$, the number of iterations in this first stage is thus given by

$$\sum_{p \neq \rho} \lfloor \log_2 \frac{|a_p|}{|a_{p^\wedge}|} \rfloor \leq (P-1) \log_2 \frac{|a_{p_1}|}{|a_{p^\wedge}|} . \tag{20}$$

Subsequently, in the second phase, the algorithm will subdivide boxes by proceeding alternatively through the variables one by one *ad infinitum*. If one or more $a_p$ is zero, the analysis above can be applied on the nonzero parameters. This reduces the number of iterations until the first phase concludes and the second starts below $\|\mathbf{a}\|_0 \log_2 \frac{\max_p |a_p|}{\min_{\{p:a_p \neq 0\}} |a_p|}$.

# B. Details of Numerical Experiments

This section gives additional details and discussion of the numerical results presented in Section 5. All tree-based models are estimated using Scikit-Learn (Pedregosa et al., 2011).

## B.1. Rotation Prediction Study Additional Details

In this study, when performing a rotation, we used only the $\sqrt{P}$ many PCA components or Active Subspace dimensions, and appended these to the original design matrix as new variables. We measure prediction error using 100-fold cross validation.

The table below gives the parameters of the datasets used for the prediction study of Section 5.2.

| Name | N | P | URL |
|------|---|---|-----|
| concrete | 1,030 | 9 | https://archive.ics.uci.edu/dataset/165/ |
| kin40k | 40,000 | 9 | https://github.com/alshedivat/keras-gp/kgp/datasets/kin40k.py |
| keggu | 65,554 | 28 | https://www.genome.jp/kegg/pathway.html |
| bike | 17,379 | 13 | https://archive.ics.uci.edu/dataset/560/ |
| obesity | 2,111 | 24 | https://archive.ics.uci.edu/dataset/544/ |
| gas | 36,733 | 12 | https://archive.ics.uci.edu/dataset/224 |
| grid | 10,000 | 13 | https://archive.ics.uci.edu/dataset/471/ |
| supercond | 21,263 | 82 | https://archive.ics.uci.edu/dataset/464/ |

We also provide boxplots of Cross Validation errors in Figure 8. Running this study took about five hours on a 40 core Ubuntu machine with 128 GB of RAM.

## B.2. Active Subspace Estimation Study Additional Details

In Section 5.3, We randomly sampled a unit vector $\mathbf{a}$ from the uniform distribution over directions and then sampled input points uniformly at random on the unit cube. We subsequently evaluated the function $f(\mathbf{x}) = \cos(6\pi(\mathbf{a}^\top(\mathbf{x} - 0.5))$ which was treated as the noiseless observed response $\mathbf{y}$. We compared the estimates with using the angle each made with $\mathbf{a}$. This experiment was repeated 20 times.

We used the implementation of PRA provided with the pypi package PSDR[1]. For GP-based active subspace estimation, we used the CRAN package activegp. We used all default settings for the PRA method as well as for the GP method. For the DASM, we used a neural network with an additional layer of width 512 subsequent to the active subspace layer, and used gradient descent with a step size of $10^{-3}$ on the Mean Squared Error cost function. This neural network was implemented in JAX (Bradbury et al., 2018). In addition to the quantitative advantages enjoyed by the tree-based method of active subspace estimation, we would also like to note that like the GP-based method, and unlike the PRA and DASM, it provides an estimate of the entire active subspace matrix, rather than simply a basis for the active subspace. This is important for two reasons. First, with the active subspace matrix in hand, we can create analogs of PCA scree plots to determine what dimension of the active subspace is most desirable, or to get some idea of how much information is being lost in, say, a two dimensional visualization. And secondly, it allows us to decide on an active subspace dimension *after* having seen the data rather than before, without requiring the estimation procedure to be re-run.

Running this study took about eight hours on a 40 core Ubuntu machine with 128 GB of RAM.

## B.3. NHEFS Data Analysis Details

This table presents the first 3 eigenvectors of the mortality data analysis, restricting to the top 9 variables with highest coefficients. The first eigenvector captures almost entirely the age variable, which (Lundberg et al., 2019) also found to be most important. We see that the second eigenvector is evenly distributed across sex and one of the urine variables. The third eigenvector is dominated by the urineDark variable.

| Eigenvector | age | urineDark | sex | urineNeg | SGOT | hemoglobin | urineAlb | total | physical |
|-------------|-----|-----------|-----|----------|------|------------|----------|-------|----------|
| 1 | -1.00 | -0.00 | 0.01 | 0.01 | -0.00 | -0.00 | 0.00 | -0.00 | -0.00 |
| 2 | 0.01 | -0.15 | 0.56 | 0.53 | -0.28 | -0.30 | 0.30 | -0.16 | 0.09 |
| 3 | -0.00 | -0.96 | -0.03 | -0.11 | 0.16 | 0.07 | 0.02 | 0.04 | 0.07 |

---

[1]https://psdr.readthedocs.io/en/latest/

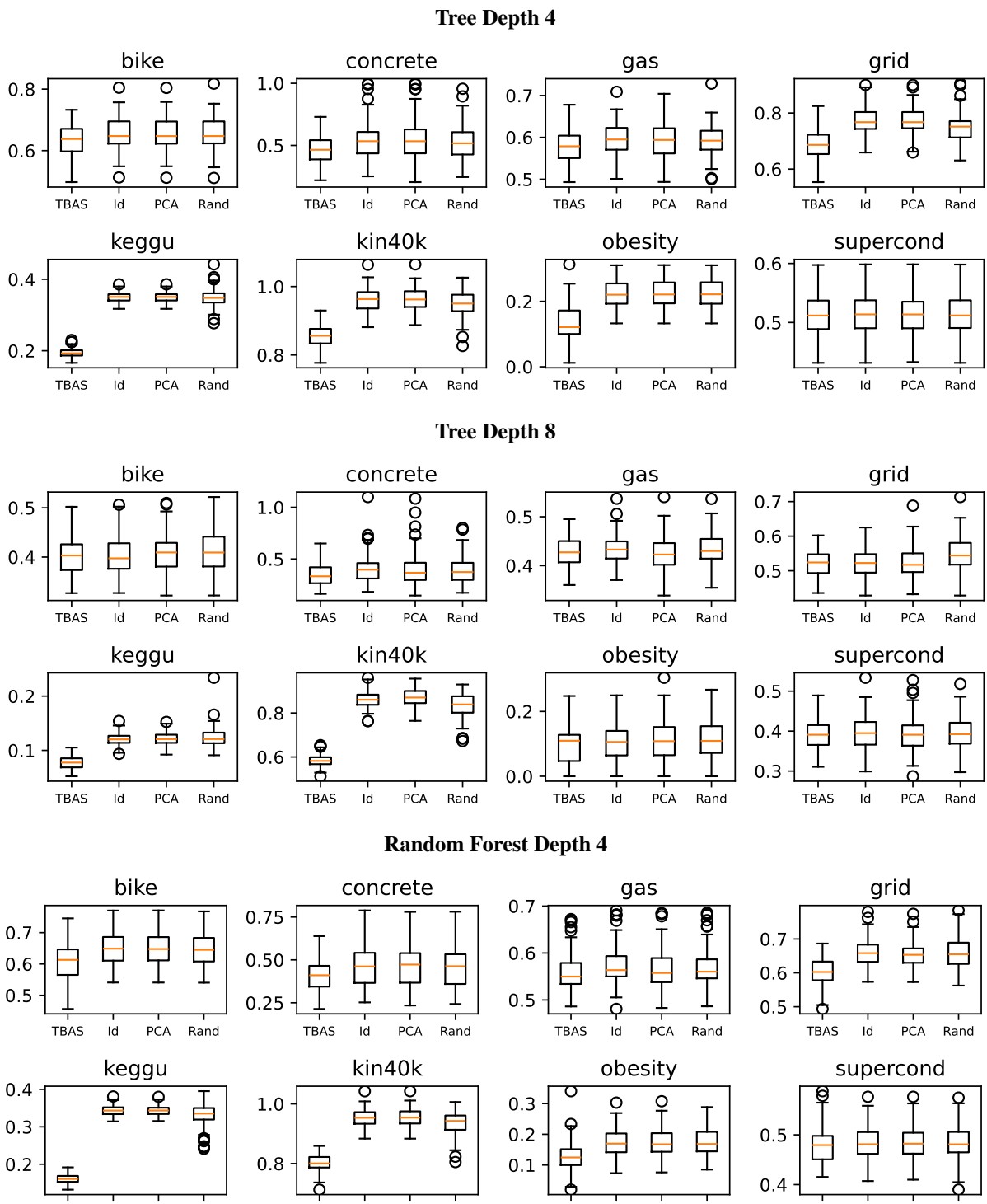

*Figure 8.* Boxplots corresponding to Table 1.

When producing the right panel of Figure 7, we used the prediction after accounting for the effect of age in order to demonstrate the change in the predictive surface over the second and third eigenvalues.

## C. Numerical Experiments on Classification Trees

We repeat the experiments of Section 5.2, but now by replacing each regression problem with a classification problem by assessing whether a given observation falls above or below the median observation. The results are given in Figure 9. Intriguingly, the results are significantly less promising for the TBAS method, despite the fact that by construction, there is structure in the data that TBAS could possibly exploit. This indicates that there may be special considerations to be taken in deploying this methodology to classification problems.

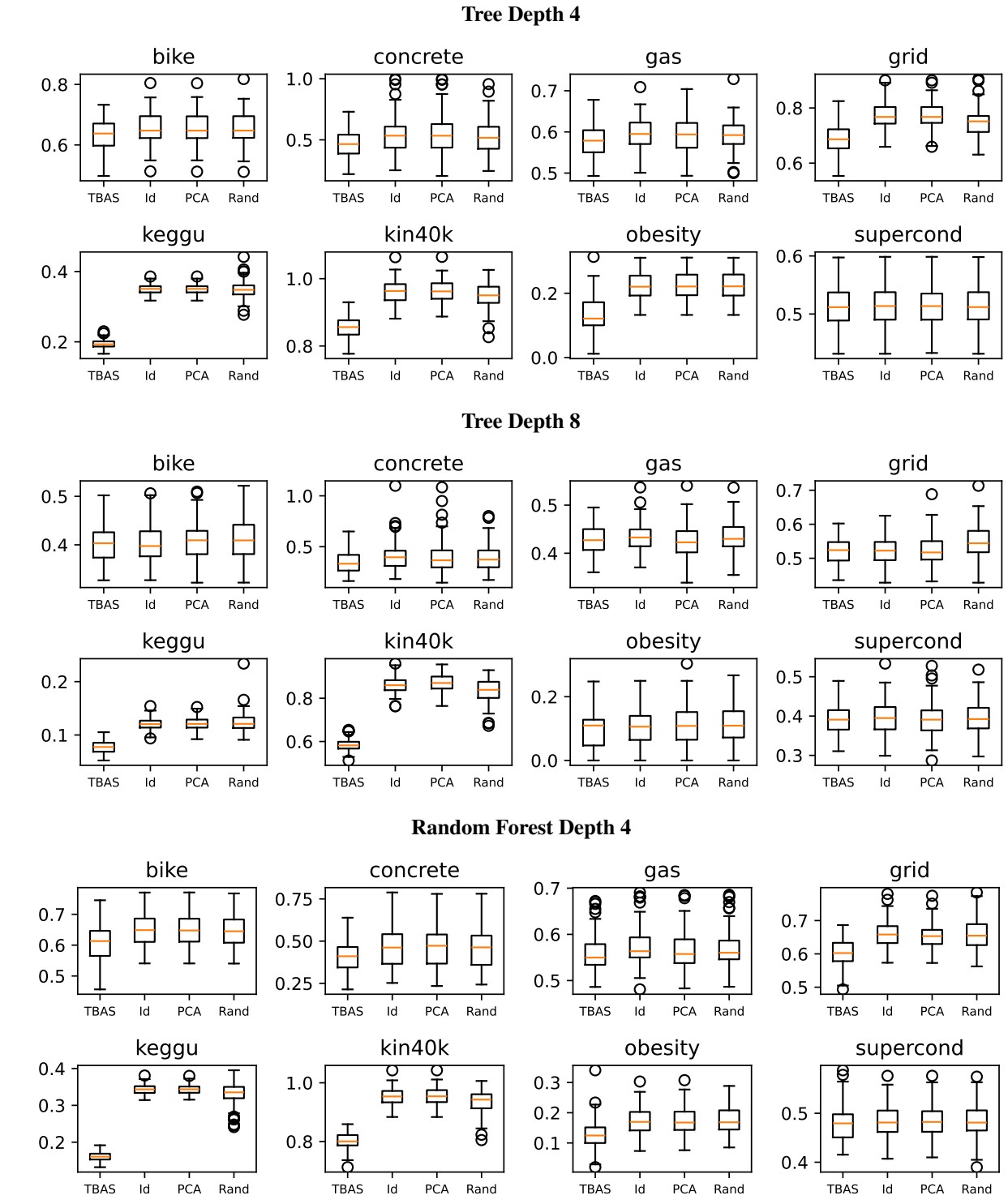

*Figure 9.* Classification Exercise: Brier Scores are indicated; lower is better.

