# OpenReview forum: "Regression Trees Know Calculus"
_ICML.cc/2025/Conference — Submitted to ICML 2025_

### Official Review · Reviewer_R71A · 2025-03-12

**Overall Recommendation:** 3

**Summary:**

The paper proposes a method to obtain gradients from regression trees. The gradient estimate is similar to a finite difference using mean responses across splits divided by size of node along dimension. Paper presents a Monte Carlo estimator and a partition-based estimator of integrated gradient quantities. Paper presents convergence of the gradient and integrated gradient estimators. Experiments include visualization of integrated gradient for MNIST digit classification, prediction error on rotated feature matrices, active subspace error compared with other methods, and dimension reduction using active subspace.

**Claims And Evidence:**

claim is that paper developed a gradient estimator (and integrated gradient estimator) for regression trees. This claim has qualitative evidence from Fig. 1. The proofs provide theory that estimators converge to true values.

Claim is that gradient of regression trees is useful/performs better than other methods for integrated gradients and active subspace. The tree-based active subspace (TBAS) rotation augmented regression had lower or equal error compared with other methods. The TBAS had lower error or execution time compared with other methods (fig 5, 6). TBAS can be used for dimension reduction / interpretability (fig 7, 4).

**Essential References Not Discussed:**

not familiar enough with area to know

**Experimental Designs Or Analyses:**

see methods/evaluation section.

The experiments presented do not include a quantitative verification of the asymptotic correctness of gradient / integrated gradient estimate. Paper states a limitation is very deep regression tree may be required for high-dimensional dense gradient. Paper is missing the experimental analysis of depth of gradient required, and the effect of relevant factors (total data, number of dimension, density of gradient etc).

**Methods And Evaluation Criteria:**

Integrated gradient for trees included qualitative evaluation of MNIST digits.

In the active-subspace rotation experiment, paper compares the RMSE of regression tree or random forest trained on augmented rotation data by TBAS, PCA, and random orthogonal directions. This seems reasonable.

Paper claims that tree-based active subspace estimation is faster and lower error than other methods (DASM, GP, PRA). Evaluation is a comparison of error and execution time. This part would benefit from complexity analysis if available.

Dimension reduction using TBAS was evaluated qualitatively, with the first most important variable matching one other study.

**Other Comments Or Suggestions:**

line 270 "we begin with MCE", then following line is about PBE.

line 273 typo.

line 368 typo.

line 348 typo.

typo in Fig 7 caption.

Fig 8 and 9 are same figure.

**Other Strengths And Weaknesses:**

Strengths: The concept of the paper (gradients for tree models) seems original. There is a good amount of comparison with existing methods in the experiments section.

Weaknesses: Claims of computational efficiency are less convincing. Paper could be improved by including theoretical complexity across methods, including proposed method. Paper states a limitation is very deep regression tree may be required for high-dimensional dense gradient. Would be good to see quantitative and empirical analysis of this.

**Questions For Authors:**

1. What is the color representing in Fig. 7? What is the takeaway from Fig.7 right panel?

2. What is the x axis representing in Fig. 6?

3. On Fig. 4, why is IG only shown on the dark pixels of digits? Is the IG value low for everything not shown?

4. Would be good to know how parameters affect the tree based gradient estimates in the empirical setting.

**Relation To Broader Scientific Literature:**

Paper points to Chaudhuri 1995 and Low 2011 as papers looking at gradients for tree based models. Paper fits in with literature on integrated gradient and active subspace method.

**Theoretical Claims:**

Did not look into in detail. Seems reasonable since estimators match intuition for finite difference.

---

> ### Author Rebuttal · Authors · 2025-04-01
>
> Thanks much for your helpful comments. In addition to our responses to your helpful feedback and questions, we have conducted a new simulation study investigating the empirical performance of our method in estimating gradients (see below and Figure 1 [HERE](https://imgur.com/a/icml-11817-rebuttal-yyVmZpe)).
>
> **Regarding: "Claims of computational efficiency are less convincing. Paper could be improved by including theoretical complexity across methods, including proposed method."**
>
> Great idea: mathematical complexity is a great starting point; we comment on it below and will add this discussion to the paper. However, it is somewhat limiting in this context with competitors defined via nonlinear optimization, as efficiency depends on how many training iterations are necessary. While we can compute the per-iteration complexity, this may not be reflective of the actual performance. This is what motivated us to use Wall time to measure computational efficiency in Figure 5 of the original article, which showed that, empirically, the tree-based methods are considerably more 1-2 orders of magnitude more efficient than the comparators.
>
> We now give theoretical complexity estimates for each method. We have added this discussion to the article.
>
> i) Regression Tree: Fitting a tree is of complexity $PN\log N$ (e.g. Chen and Guestrin 2015's xgboost paper). Then, computing the gradient estimates requires a fixed number of computations at each decision node in the tree, of which there are on the order of $N\log N$. Storage is $P N\log N$ as each node has a gradient estimate.
>
> ii) Gaussian process: Evaluating the likelihood is of complexity $N^3P$. The number of iterations required is dependent on the exact sampling scheme and function, and not well understood. After fitting the model, the active subspace can be extracted with complexity $N^2P^2$.
>
> iii) Polynomial Ridge Approximant: This is solved via a nonlinear least squares problem, and each iteration will have complexity $NPR$ where $R$ is the active subspace dimension. Like GPs, the number of iterations needed to converge is difficult to do analysis on. However, once the analysis is done, the active subspace matrix is immediately available with no further computations.
>
> iv) Deep Active Subspace. This involves fitting a neural network. The per-iteration complexity scales with $MP$, where $M$ is the minibatch size and $P$ is the input dimension, and this value is scaled by the time it takes to do a forward pass through the network. Like the GP and PRA, the exact number of iterations required is difficult to know. Also like the PRA, the first weight matrix encodes the active subspace.
>
> **Paper states a limitation is very deep regression tree may be required for high-dimensional dense gradient. Would be good to see quantitative and empirical analysis of this.**
>
> Thanks for suggesting that we look further into this. We have conducted a new simulation study investigating the impact of tree depth on Gradient performance on the function $f(x) = \log(1+a^\top x/P)$ with nonzero elements of a generated from an iid Gaussian, which shows the convergence in finite samples explicitly. See Figure 1 [HERE](https://imgur.com/a/icml-11817-rebuttal-yyVmZpe) for results. The left column shows the performance when the Depth=4, and the right column for Depth=12. The x axis in each pane gives sample size and the y axis gives dimension. We see that when Depth=4, there is essentially no reduction in error. By contrast, when Depth=12, the error decreases to zero with larger sample size, illustrating the importance of depth. We have added this experiment to Section 5.
>
> **Figure 8 and 9 are same figure.**
> Thanks so much for pointing this out; Figure 9 was supposed to give the boxplots for classification, but we accidentally replicated the regression results because the filenames were similar. We have fixed this.
>
> **"What is the color representing in Fig. 7? What is the takeaway from Fig.7 right panel?"**
> Thanks for pointing out that we did not mention that the color indicates the predicted value; pink is low and blue is high. The take-away from the right panel is that the predictive surface is actually quite simple when viewed via the active subspace, and is almost like an "XOR" shape.
>
> **"What is the x axis representing in Fig. 6?"**
> x-axis is sample size; y-axis is angle between true and estimated subspace.
>
> **On Fig. 4, why is IG only shown on the dark pixels of digits? Is the IG value low for everything not shown?**
> Since the reference image is all white, any white pixel in a target image will have IG 0.
>
>
> **Would be good to know how parameters affect the tree based gradient estimates in the empirical setting.**
> See discussion of the additional new Figure 1 above.
>
>
> Thanks for your suggestions that we look into the computational efficiency and depth requirements; we think addressing these concerns has much improved the article.

---

### Official Review · Reviewer_3jeE · 2025-03-14

**Overall Recommendation:** 4

**Summary:**

The paper develops a simple and computationally efficient approach for estimating gradients from a decision tree, essentially by computing a finite difference across all of the nodes on the way to the leaf that contains the point at which a gradient is required. These gradients are then used for active subspace estimation and computing integrated gradients.

## update after rebuttal
Score up to Accept

**Claims And Evidence:**

The paper provides a thorough theoretical analysis to support what on the surface seems a fairly natural and straightforward approach to estimating gradients from a decision tree. Overall I felt like the claims made by the paper are well-supported, with some qualifications below.

**Essential References Not Discussed:**

Not that I'm aware of.

**Experimental Designs Or Analyses:**

I was confused by the use of random forests in 5.1. It seems that the gradients are themselves being computed on the random forest model? Is that as the average of the gradients across the forest or something? Or is a separate regression tree being used as an explanation model?

Generally, the empirical evaluation covers two things: integrated gradients for model explanation, and active subspace estimation for rotation or dimensionality reduction. I didn't see any issues with the experimental design related to active subspace estimation. For integrated gradients, the evaluation is much weaker, and in fact is not really an evaluation but rather an illustration, lacking comparison methods or ground truth. (The paper describes it as a "qualitative study," which is somewhat euphemistic).

There is a third empirical evaluation that I think would be important for a paper like this but doesn't seem to be present: evaluation of the actual quality of the derivative estimates. A major claim of the paper is that regression trees can be used for uncertainty quantification, displacing models like GPs that are usually used for estimating gradients of black-box functions. But all of the evaluation is on downstream uses of gradients. How about an evaluation just of how the gradients compare to ground truth, compared to a GP estimate of the gradients? There is a large number of smooth benchmark problems used for global sensitivity analysis that would be suitable for this type of experiment. This is a pretty major hole in the paper I think, as I don't feel confident that the method is necessarily computing gradients as well as a GP on low- or mid-dimensional problems with dense gradients (a distinct task from active subspace estimation, but vital for many UQ problems).

**Methods And Evaluation Criteria:**

The use of active subspace as a way of evaluating the approach is great, and shows a real use-case. Derivative-based sensitivity analysis is another one that the authors could consider for the future, and where there are standardized benchmark problems and existing (GP-based usually) baselines.

**Other Comments Or Suggestions:**

typo at the bottom of page 4, "nod x"
bottom of page 5, "fit the to two subsets"

**Other Strengths And Weaknesses:**

Overall the paper is very well-written and presents what may be a useful method.

Is it true that decision trees are a workhorse of the contemporary data scientist? I see this as true via their use as a component in Random Forests and XGBoost, which I would certainly agree are workhorses of the contemporary data scientist. The paper would be strengthened by some analysis of how the method can be used together with Random Forests in particular. The obvious thing would be average gradients across the forest, but the paper describes deep trees as being important for estimating gradients, while Random Forests are usually ensembles of shallow trees. The authors thoughts on this question would be very helpful.

**Questions For Authors:**

* Can you provide evaluation of how accurately the derivatives are estimated in low- to mid-dimensional problems with dense gradients, compared to a GP and ground truth?

* Can this method be used together with a random forest?

**Relation To Broader Scientific Literature:**

The relation to broader scientific literature seems OK to me.

**Theoretical Claims:**

Not in detail.

---

> ### Author Rebuttal · Authors · 2025-04-01
>
> Thanks much for your helpful comments. In addition to our responses to your feedback and questions, we have conducted two new simulation studies, 1) investigating the empirical performance of our method relative to Gaussian Processes in estimating gradients (see below and Figure 2 [HERE](https://imgur.com/a/icml-11817-rebuttal-yyVmZpe)), and 2)  investigating the empirical accuracy of our method (see below and Figure 1 [HERE](https://imgur.com/a/icml-11817-rebuttal-yyVmZpe))
>
>
> **Regarding: “I was confused by the use of random forests in 5.1. It seems that the gradients are themselves being computed on the random forest model? Is that as the average of the gradients across the forest or something? Or is a separate regression tree being used as an explanation model?”**
>
> Thanks for pointing out that this was not well-explained; we indeed used an average of estimates from each tree within the forest as our estimator for the whole forest. We have added a discussion of this to Section 5.2.
>
> **Regarding: "There is a third empirical evaluation that I think would be important for a paper like this but doesn't seem to be present: evaluation of the actual quality of the derivative estimates. A major claim of the paper is that regression trees can be used for uncertainty quantification, displacing models like GPs that are usually used for estimating gradients of black-box functions. But all of the evaluation is on downstream uses of gradients. How about an evaluation just of how the gradients compare to ground truth, compared to a GP estimate of the gradients? There is a large number of smooth benchmark problems used for global sensitivity analysis that would be suitable for this type of experiment. This is a pretty major hole in the paper I think, as I don't feel confident that the method is necessarily computing gradients as well as a GP on low- or mid-dimensional problems with dense gradients (a distinct task from active subspace estimation, but vital for many UQ problems)."**
>
> Thanks for this comment; we agree that it’s helpful to have a direct evaluation of the gradient estimate quality. We have implemented a new simulation showing the empirical performance of TBGE in estimating gradients under various tree depth, sample sizes, dimensions and gradient densities on the function $f(x) = \log(1+a^\top x/P)$ with nonzero elements of a generated from an iid Gaussian, which shows the convergence in finite samples explicitly. See Figure 1 [HERE](https://imgur.com/a/icml-11817-rebuttal-yyVmZpe) for results. We have added this experiment to Section 5.
>
> Regarding the comparison to GPs: we certainly don’t think that tree-based methods will displace GPs for gradient estimation on all problems. In small dimension with smooth functions, a dense gradient and a small sample size, a GP will greatly outperform the TBGE. In fact, for a sufficiently smooth function, we suspect a GP will basically always outperform a TBGE for a fixed sample size. However, the relative scalability of regression trees compared to GPs means they may be more attractive in large sample settings, even for dense estimators in lower dimension, particularly under noise and when speed is more important than efficiency. To illustrate what we mean, we consider the problem where the simulator of interest is relatively cheap to evaluate, and we have a large number of samples which we wish to use to estimate the gradient in several locations in the input space. We have conducted three simulation studies estimating gradients on the Levy, Cosine Ridge function, and Ackley function with variable sample size and iid Gaussian noise with standard deviation 0.1 (results in Figure 2 [HERE](https://imgur.com/a/icml-11817-rebuttal-yyVmZpe)). The x-axis shows not sample size, but the elapsed real time. We see that using a regression tree on a larger dataset can give a better time-accuracy trade-off for lower accuracies than a GP on a smaller dataset. Additionally, on the Ackley function, the function is sufficiently rough that the GP often treats that variability as noise, and is unable to effectively form gradient estimates, whereas the regression tree is able to.
>
> Certainly, we don’t mean to suggest that analysts can throw away all their existing methods for gradient estimation; but on problems in moderate dimension or with low effective dimension and many datapoints, we found TBGE to perform well, and we think this is a really interesting finding given how small a roll regression trees play in the UQ space today.
>
> Thanks very much for challenging us to consider these new aspects of our method; we think that addressing them has improved and clarified the article.

---

> > ### Comment · Reviewer_3jeE · 2025-04-04
> >
> > Thank you for the new results and additional analysis. I think the paper is strengthened by including some more detail on the situations in which this approach should be used vs. GP. In any case, there certainly are situations in which this will be a useful tool.

---

### Official Review · Reviewer_iY41 · 2025-03-14

**Overall Recommendation:** 4

**Summary:**

The paper proposes an estimator of gradients and integrated gradients based on regression trees. The proposed method estimates function gradients by finite differences between adjacent regions split by a regression tree node. Building upon this estimator, Monte Carlo based and partition-based estimators are developed to estimate integrated gradients. Theoretical guarantees of the estimation consistency are also developed in the paper. The proposed method is then applied to active subspace methods for dimension reduction and integrated gradient methods for model interpretation.

## update after rebuttal

I thank the author(s) for their detailed rebuttal and additional experiment results. Most of my concerns are resolved and I am raising my score to 4.

**Claims And Evidence:**

The claims in the paper are in general grounded. With that being said, the paper misses some important aspects in methodology and experiments, making the results not entirely convincing. Please see my comments in the "Methods And Evaluation Criteria" section and the "Experimental Designs Or Analyses" section.

**Essential References Not Discussed:**

N/A

**Experimental Designs Or Analyses:**

The experiments are well-designed to cover different use cases. However, some aspects are not assessed or discussed in the experiments:
* While Theorem 4.1 establishes the large sample property of the proposed gradient estimator, what would be its empirical performance with finite sample size?
* Related to my previous comment on correlated covariates, how would this affect the performance of gradient estimation and active subspace discovery?
* The simulation experiments in Sections 5.3 and 5.4 only consider noiseless observations. It would be interesting to see how the active subspace estimation performance under different noise levels.
* It would be interesting to compare the empirical performance of Monte Carlo based and partition-based integral estimation.
* In the experiments, what regression tree hyperparameters did you use to fit TBAS?

**Methods And Evaluation Criteria:**

Overall, the proposed methodology in the paper is technically sound. Some clarification on the following questions would be appreciated:
* My understanding is that the estimators rely on a sufficiently deep regression tree. In practice, how should one tune the depth of the tree when using, e.g., TBAS?
* Instead of using a single deep tree, would a tree ensemble help with the estimation?
* The gradient estimators rely on the splits on each covariate, but this could be in trouble when the covariates are highly correlated, since the tree may always split on one covariate but never split on the other. How would the proposed method handle such scenarios?

**Other Comments Or Suggestions:**

The paper is well-written, except for some minor issues:
* The notations for $u$ and $l$ in Algorithm 1 are different from the ones used in Equation (3).
* Line 207, LHS: "We begin with the MCE" should be "We begin with the PB".
* The Appendix needs careful proofreading. For instance,
	* Line 552: Is Theorem A actually Theorem 4.1?
	* Line 612: Is Proposition 1 actually Theorem 4.1?
	* Lines 605 and 644: "Under Assumptions A and A"

**Other Strengths And Weaknesses:**

Please see my comments in "Relation To Broader Scientific Literature" for the assessment of the paper's novelty and contribution.

**Questions For Authors:**

N/A

**Relation To Broader Scientific Literature:**

The paper presents an interesting and novel decision tree based methods for estimating gradients and integrated gradients, which provides a promising alternative to existing active subspace estimation methods in my opinion.

**Theoretical Claims:**

The theoretical results in the paper are stated in a rigorous manner, except for Theorem A.1, where it is unclear whether the result is almost sure convergence or convergence in probability.

I briefly reviewed the proofs and they look reasonable to me, though I did not carefully examine them line by line.

---

> ### Author Rebuttal · Authors · 2025-04-01
>
> Thanks much for your helpful comments! In addition to our responses to your feedback and questions, we have conducted three new simulation studies 1) investigating the effect of correlation on our gradient estimates (see below and Figure 6 [HERE](https://imgur.com/a/icml-11817-rebuttal-yyVmZpe)), 2) investigating the empirical performance of our method in estimating gradients (see below and Figure 1 [HERE](https://imgur.com/a/icml-11817-rebuttal-yyVmZpe)), and 3) Investigating the impact of noise on the active subspace estimation capability (see below and Figure 3 [HERE](https://imgur.com/a/icml-11817-rebuttal-yyVmZpe)).
>
> Forgive us for being terse in some parts of this rebuttal; we ran up against the character space constraints.
>
> **In practice, how should one tune the depth of the tree when using, e.g., TBAS?**
>
> This is an interesting question. Our theoretical analysis suggests that the number of observations per leaf node should scale with P Log(N), and this seems to be a good default. However, it would be interesting to investigate whether it is sufficient to just tune the regression tree for predictive purposes; recent work [1](https://arxiv.org/abs/2208.10664) has found this to be the case in spline-based gradient estimation, and it’s possible that this is the case for trees as well. We think this deserves an article of its own to investigate fully.
>
> **Regarding Tree Ensembles**
>
> We realize the article is not clear as it stands: we do in fact perform numerical experiments which average active subspace estimates from different regression trees within a random forest. We do this via simple averaging of the estimated active subspaces, which as an average of consistent estimators will also be consistent. We have clarified this in the text.
>
> **Regarding the effect of Correlation**
>
> We conducted a new simulation study to investigate this interesting point (see Figure 6 [HERE](https://imgur.com/a/icml-11817-rebuttal-yyVmZpe)). Sampling data from a truncated normal in 5D with correlation varying from 0 to 0.99, we evaluated estimates of the Ackley function's gradient at 100 random points. We can confirm the effect that you have conjectured exists; intriguingly, however, it requires a very high level of correlation before the estimates are significantly affected.
>
>
> **"The theoretical results in the paper are stated in a rigorous manner, except for Theorem A.1, where it is unclear whether the result is almost sure convergence or convergence in probability"**
>
> Thanks very much for pointing out that we did not specify the convergence mode; we have clarified that we meant convergence in probability.
>
> **Regarding empirical performance with finite sample size**
>
> Thanks for this question; we have implemented a new simulation showing the empirical performance of TBGE in estimating gradients under various tree depth, sample sizes, dimensions and gradient densities on the function $f(x) = \log(1+a^\top x/P)$ with nonzero elements of $a$ generated from an iid Gaussian, which shows the convergence in finite samples explicitly. See Figure 1 [HERE](https://imgur.com/a/icml-11817-rebuttal-yyVmZpe) for results.
>
> **Regarding effect of noise in active subspace estimation**
> Thanks for this suggestion; we have reran the experiments in Figure 5 of the article/Section 5.3 with iid normal noise with a standard deviation of 0.1. The results are in Figure 3 [HERE](https://imgur.com/a/icml-11817-rebuttal-yyVmZpe). The comparative performances are quite similar, though the TBAS even better in dimensions 4 and 5.
>
> **"It would be interesting to compare the empirical performance of Monte Carlo based and partition-based integral estimation".**
> Thanks for this suggestion; it is always interesting to consider additional experiments, but in this case, since the Monte Carlo method converges to the partition-based method in the limit of increasing Monte Carlo sample size, we suspect that the partition-based approach would be superior.
>
> **"In the experiments, what regression tree hyperparameters did you use to fit TBAS?"**
> Thanks for pointing out that this information was missing from our appendix; we mostly used scikit-learn defaults and have added the following to Appendix B:
> “””
> We used the DecisionTreeRegressor and RandomForestRegressor models from scikit-learn using default parameters together with a tree depth of 4 or 8 and a minimum number of observations per node of 5.
> “””
>
> **Regarding Other Comments Or Suggestions:**
> Thanks for catching the errors regarding l and u, for the “we begin with MCE” mistake, and for pointing out the labeling issue in the Appendix; we have resolved all of these issues.
>
> Thanks very much for your thoughtful comments; you brought to light some interesting issues we hadn’t considered and we think the paper is more comprehensive now that it discusses them.

---

### Official Review · Reviewer_98rh · 2025-03-14

**Overall Recommendation:** 3

**Summary:**

In this paper, the authors propose an efficient method to estimate the gradients of the underlying function learned by regression trees. In a nutshell, by computing a quantity resembling finite differences at a tree’s nodes—based on the extent of a given node and the function values in its subtrees—one can estimate different entries of the gradient. This allows for efficient gradient computation by simply traversing the tree and computing these values for each node. To evaluate the quality of the proposed gradient estimation method, the authors apply it in the context of model interpretability and performance improvements, specifically through Integrated Gradients, Active Subspace Estimation, and dimensionality reduction. The method outperforms existing approaches in terms of both computational complexity and predictive performance.

**Claims And Evidence:**

One of the major claims of the paper is that the gradient estimation obtained through the proposed procedure is accurate. The authors support this claim with both theoretical analysis and experimental validation, which adequately substantiate their argument.

**Essential References Not Discussed:**

No, there are no critical references missing in the paper.

**Experimental Designs Or Analyses:**

The experiments primarily use small-scale datasets, limiting the method’s generalizability. Evaluating it on larger datasets, such as (1) Criteo, (2) NYC Taxi, and (3) Higgs Boson, would better assess its scalability and robustness, strengthening the paper’s claims.

**Methods And Evaluation Criteria:**

The authors validate their method on several popular tabular datasets (UCI datasets) and an image dataset (MNIST), providing insights into the effectiveness of the approach from different perspectives. However, the considered datasets are relatively small in scale (thousands of samples). Demonstrating results on larger datasets, such as (1) Criteo Conversion Log Dataset, (2) NYC Taxi Trip Duration, or (3) Higgs Boson Challenge, would significantly strengthen the paper. While it is understandable that tree-based models have their limitations, the current experiments may not be sufficient to draw confident conclusions about the method’s scalability and generalization.

**Other Comments Or Suggestions:**

* Figure 4: not very clear, would be good to visually highlight each pair
* Figure 7: twice "Left"

**Other Strengths And Weaknesses:**

Strengths:

* The method is highly efficient and demonstrates strong gradient estimation quality across several interesting applications.

Weaknesses:

* The generalizability of the current experimental results is a limitation. Extending the evaluation to larger datasets could significantly strengthen the contribution.

**Questions For Authors:**

* How is the gradient computed when multiple nodes in a tree use the same feature for splitting? Is any weighting applied?

**Relation To Broader Scientific Literature:**

The paper clearly positions itself within the existing literature by thoroughly discussing prior work. It provides sufficient detail on potential applications of estimated gradients, such as gradient-based model interpretation, while highlighting the limitations of existing methods. More broadly, the authors present their approach as the first effective and efficient method for gradient estimation in tree-based models.

**Theoretical Claims:**

The main theoretical contributions of the paper are Theorems 4.1 and 4.2, which establish that the estimated gradient-based quantities (the gradient itself and integro-differential quantities) are close to their true values. The proposed proofs appear to be correct and valid, with no observable issues.

---

> ### Author Rebuttal · Authors · 2025-04-01
>
> Thanks much for your helpful comments! We have incorporated more datasets as suggested. Below, we discuss these new results and subsequently respond to your other helpful feedback.
>
> **Regarding your suggestions for more datasets:**
>
> Thanks for suggesting these datasets. We have incorporated the Taxi Trip Duration and Higgs Boson datasets into our experiments (see below), and we think they are both very interesting. However, regarding the Criteo dataset, perhaps it is simply because we are not familiar with this dataset, but it seems like the predictor variables are primarily categorical. As such, we are not usefully able to define derivatives for them and we did not include this dataset.
>
> The Higgs Boson dataset seems perfect to demonstrate the advantage of our method, with a moderate dimension and huge dataset size. Indeed, our TBAS significantly outperforms the other transformations on this dataset; and did not misclassify a single observation in our experiments, leading to a 0% error rate (the other methods are around 1%; see Figure 4 [HERE](https://imgur.com/a/icml-11817-rebuttal-yyVmZpe); RMSE on a classification problem gives Brier Score). The taxi dataset is somewhat less within the optimal target application, having just a few variables, namely latitude, longitude and time for pickup and delivery, together with number of passengers. Feature engineering is presumably key to making good progress on this problem, and we are not time-series experts.  Nevertheless, the TBAS still outperforms the identity and random transformation. However, the PCA-based method totally outperforms all other approaches on this dataset! We were very surprised by this given how low dimensional the problem was (i.e. we didn’t expect any transform to have much of an effect) and don’t yet have an explanation for this phenomenon. See again Figure 4 [HERE](https://imgur.com/a/icml-11817-rebuttal-yyVmZpe). We will fold these results into Tables 1 and Figures 8 and 9 of the article.
>
> **Question: How is the gradient computed when multiple nodes in a tree use the same feature for splitting? Is any weighting applied?**
>
> In this article, we simply “overwrote” the previous estimate of the partial derivative from the node earlier in the tree with the later node. We think that using some kind of weighted combination between the two could indeed yield better performance, but significant though would have to go into this to determine what the optimal trade-off is between using the coarser estimate with more data (but which is less localized) and the finer, localized estimate with less data. This would depend probably on something like the local Lipschitz constant of the function, and we think there is significant analysis and computational experiments needed to determine the best way to do this.
>
> **Regarding the “Other comments”:**
> Thanks for very much for pointing out the error in Figure 7 (it’s been corrected); thanks for pointing out that Fig 4 was unclear; we have added lines between each pair to make it clearer, see Figure 5 [HERE](https://imgur.com/a/icml-11817-rebuttal-yyVmZpe).
>
> We think the paper is already much improved by your suggestions and especially with the inclusion of these new datasets; thanks again for your helpful review.

---

### Decision · Program_Chairs · 2025-05-01

**Decision:**

Reject

**Comment:**

The paper presents a way to estimate gradients for functions fitted by trees. While the reviewers and I agree that this is a solid contribution, I believe the paper would be greatly strengthened by including results on even more and larger datasets (as suggested by 98rh); datasets with correlated covariates (see reply to iY41); and from the empirical evaluation suggested by 3jeE. I also believe that aditional empirical investigation into tree depth (see review by IY41) would make for a stronger paper.